# UniKGQA: Unified Retrieval and Reasoning for Solving Multi-hop Question Answering over Knowledge Graph

**Jinhao Jiang**[1,3][*]**, Kun Zhou**[2,3][*]**, Wayne Xin Zhao**[1,3][✉] **and Ji-Rong Wen**[1,2,3]
[1]Gaoling School of Artificial Intelligence, Renmin University of China.
[2]School of Information, Renmin University of China.
[3]Beijing Key Laboratory of Big Data Management and Analysis Methods.
`jiangjinhao@ruc.edu.cn, francis_kun_zhou@163.com,`
`batmanfly@gmail.com, jrwen@ruc.edu.cn`

## ABSTRACT

Multi-hop Question Answering over Knowledge Graph (KGQA) aims to find the answer entities that are multiple hops away from the topic entities mentioned in a natural language question on a large-scale Knowledge Graph (KG). To cope with the vast search space, existing work usually adopts a two-stage approach: it first retrieves a relatively small subgraph related to the question and then performs the reasoning on the subgraph to find the answer entities accurately. Although these two stages are highly related, previous work employs very different technical solutions for developing the retrieval and reasoning models, neglecting their relatedness in task essence. In this paper, we propose UniKGQA, a novel approach for multi-hop KGQA task, by unifying retrieval and reasoning in both model architecture and parameter learning. For model architecture, UniKGQA consists of a semantic matching module based on a pre-trained language model (PLM) for question-relation semantic matching, and a matching information propagation module to propagate the matching information along the directed edges on KGs. For parameter learning, we design a shared pre-training task based on question-relation matching for both retrieval and reasoning models, and then propose retrieval- and reasoning-oriented fine-tuning strategies. Compared with previous studies, our approach is more unified, tightly relating the retrieval and reasoning stages. Extensive experiments on three benchmark datasets have demonstrated the effectiveness of our method on the multi-hop KGQA task. Our codes and data are publicly available at `https://github.com/RUCAIBox/UniKGQA`.

## 1 INTRODUCTION

With the availability of large-scale knowledge graphs (KGs), such as Freebase (Bollacker et al., 2008) and Wikidata (Tanon et al., 2016), knowledge graph question answering (KGQA) has become an important research topic that aims to find the answer entities of natural language questions from KGs. Recent studies (Lan et al., 2021) mainly focus on *multi-hop KGQA*, a more complex scenario where sophisticated multi-hop reasoning over edges (or relations) is required to infer the correct answer on the KG. We show an example in Figure 1(a). Given the question "*Who is the wife of the nominee for The Jeff Probst Show*", the task goal is to find a reasoning path from the topic entity "*The Jeff Probst Show*" to the answer entities "*Shelley Wright*" and "*Lisa Ann Russell*".

Faced with the vast search space in large-scale KGs, previous work (Sun et al., 2018; 2019) typically adopts a *retrieval*-then-*reasoning* approach, to achieve a good trade-off. Generally, the retrieval stage aims to extract relevant triples from the large-scale KG to compose a relatively smaller question-relevant subgraph, while the reasoning stage focuses on accurately finding the answer entities from the retrieved subgraph. Although the purposes of the two stages are different, both stages

---

[*] Equal contribution.
[✉] Corresponding author.

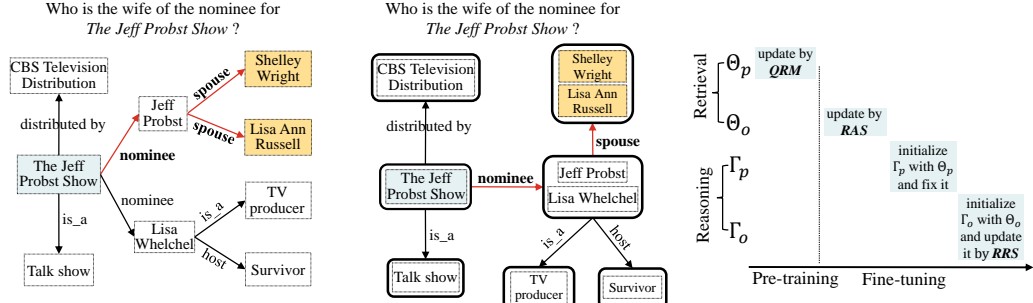

(a) an example of multi-hop KGQA    (b) an example of abstract subgraph    (c) the overall learning procedure

Figure 1: Illustrative examples and learning procedure of our work.

need to evaluate the semantic relevance of a candidate entity with respect to the question (for removal or reranking), which can be considered as a semantic matching problem in essence. For measuring the entity relevance, relation-based features, either direct relations (Miller et al., 2016) or composite relation paths (Sun et al., 2018), have been shown to be particularly useful for building the semantic matching models. As shown in Figure 1(a), given the question, it is key to identify the semantically matched relations and the composed relation path in the KG (*e.g.,* "*nominee → spouse*") for finding the correct answer entities. Since the two stages cope with different scales of search space on KGs (*e.g.,* millions *v.s.* thousands), they usually adopt specific technical solutions: the former prefers more efficient methods focusing on the recall performance (Sun et al., 2018), while the latter prefers more capable methods for modeling fined-grained matching signals (He et al., 2021).

Considering the same essence for both stages, this work aims to push forwards the research on multi-hop KGQA by investigating the following problem: can we design a unified model architecture for both stages to derive a better performance? To develop a unified model architecture for multi-hop KGQA, a major merit is that we can tightly relate the two stages and enhance the sharing of the relevance information. Although the two stages are highly related, previous studies usually treat them *separately* in model learning: only the retrieved triples are passed from the retrieval stage to the reasoning stage, while the rest of the useful signal for semantic matching has been neglected in the pipeline framework. Such an approach is likely to lead to sub-optimal or inferior performance, since multi-hop KGQA is a very challenging task, requiring elaborate solutions that sufficiently leverage various kinds of relevance information from the two stages.

However, there are two major issues about developing a unified model architecture for multi-hop KGQA: (1) How to cope with very different scales of search space for the two stages? (2) How to effectively share or transfer useful relevance information across the two stages? For the first issue, instead of letting the same model architecture directly fit very different data distributions, we propose a new subgraph form to reduce the node scale at the retrieval stage, namely *abstract subgraph* that is composed by merging the nodes with the same relations from the KG (see Figure 1(b)). For the second issue, based on the same model architecture, we design an effective learning approach for the two stages, so that we can share the same pre-trained parameters and use the learned retrieval model to initialize the reasoning model (see Figure 1(c)).

To this end, in this paper, we propose UniKGQA, a unified model for multi-hop KGQA task. Specifically, UniKGQA consists of a semantic matching module based on a PLM for question-relation semantic matching, and a matching information propagation module to propagate the matching information along the directed edges on KGs. In order to learn these parameters, we design both pre-training (*i.e.,* question-relation matching) and fine-tuning (*i.e.,* retrieval- and reasoning-oriented learning) strategies based on the unified architecture. Compared with previous work on multi-hop KQGA, our approach is more unified and simplified, tightly relating the retrieval and reasoning stages.

To our knowledge, it is the first work that unifies the retrieval and reasoning in both model architecture and learning for the multi-hop KGQA task. To evaluate our approach, we conduct extensive experiments on three benchmark datasets. On the difficult datasets, WebQSP and CWQ, we outperform existing state-of-the-art baselines by a large margin (*e.g.,* 8.1% improvement of Hits@1 on WebQSP and 2.0% improvement of Hits@1 on CWQ).

## 2 PRELIMINARY

In this section, we introduce the notations that will be used throughout the paper and then formally define the multi-hop KGQA task.

**Knowledge Graph (KG).** A knowledge graph typically consists of a set of triples, denoted by $\mathcal{G} = \{\langle e, r, e' \rangle | e, e' \in \mathcal{E}, r \in \mathcal{R}\}$, where $\mathcal{E}$ and $\mathcal{R}$ denote the entity set and relation set, respectively. A triple $\langle e, r, e' \rangle$ describes the fact that a relation $r$ exists between head entity $e$ and tail entity $e'$. Furthermore, we denote the set of neighborhood triples that an entity $e$ belongs to by $\mathcal{N}_e = \{\langle e, r, e' \rangle \in \mathcal{G}\} \cup \{\langle e', r, e \rangle \in \mathcal{G}\}$. Let $r^{-1}$ denote the inverse relation of $r$, and we can represent a triple $\langle e, r, e' \rangle$ by its inverse triple $\langle e', r^{-1}, e \rangle$. In this way, we can simplify the definition of the neighborhood triples for an entity $e$ as $\mathcal{N}_e = \{\langle e', r, e \rangle \in \mathcal{G}\}$. We further use $\boldsymbol{E} \in \mathbb{R}^{d \times |\mathcal{E}|}$ and $\boldsymbol{R} \in \mathbb{R}^{d \times |\mathcal{R}|}$ to denote the embedding matrices for entities and relations in KG, respectively.

**Multi-hop Knowledge Graph Question Answering (Multi-hop KGQA).** Given a natural language question $q$ and a KG $\mathcal{G}$, the task of KGQA aims to find answer entitie(s) to the question over the KG, denoted by the answer set $\mathcal{A}_q \in \mathcal{E}$. Following previous work (Sun et al., 2018; 2019), we assume that the entities mentioned in the question (*e.g.,* "The Jeff Probst Show" in Figure 1(a)) are marked and linked with entities on KG, namely *topic entities*, denoted as $\mathcal{T}_q \subset \mathcal{E}$. In this work, we focus on solving the *multi-hop KGQA* task where the answer entities are multiple hops away from the topic entities over the KG. Considering the trade-off between efficiency and accuracy, we follow existing work (Sun et al., 2018; 2019) that solves this task using a *retrieval*-then-*reasoning* framework. In the two-stage framework, given a question $q$ and topic entities $\mathcal{T}_q$, the retrieval model aims to retrieve a small subgraph $\mathcal{G}_q$ from the large-scale input KG $\mathcal{G}$, while the reasoning model searches answer entities $\mathcal{A}_q$ by reasoning over the retrieved subgraph $\mathcal{G}_q$.

**Abstract Subgraph.** Based on KGs, we further introduce the concept of *abstract graph*, which is derived based on the reduction from an original subgraph. Specifically, given a subgraph related to question $q$, denoted as $\mathcal{G}_q \subset \mathcal{G}$, we merge the tail entities from the triples with the same *prefix* (*i.e.,* the same head entity and relation: $\langle e, r, ? \rangle$), and then generate a corresponding abstract node $\tilde{e}$ to represent the set of tail entities, so we have $\tilde{e} = \{e' | \langle e, r, e' \rangle \in \mathcal{G}\}$. Similarly, we can also perform the same operations on the head entities. To unify the notations, we transform an original node that can't be merged into an abstract node by creating a set only including the node itself. In this way, the corresponding abstract subgraph $\mathcal{G}_q$ can be denoted as: $\widetilde{\mathcal{G}}_q = \{\langle \tilde{e}, r, \tilde{e}' \rangle | \exists e \in \tilde{e}, \exists e' \in \tilde{e}', \langle e, r, e' \rangle \in \mathcal{G}_q\}$, where each node $\tilde{e}$ is an abstract node representing a set of original nodes (one or multiple). We present illustrative examples of the original subgraph and its abstract subgraph in Figure 1(a) and Figure 1(b).

## 3 APPROACH

In this section, we present our proposed UniKGQA, which unifies the retrieval and reasoning for multi-hop KGQA. The major novelty is that we introduce a unified model architecture for both stages (Section 3.1) and design an effective learning approach involving both specific pre-training and fine-tuning strategies (Section 3.2). Next, we describe the two parts in detail.

### 3.1 UNIFIED MODEL ARCHITECTURE

We consider a general input form for both retrieval and reasoning, and develop the base architecture by integrating two major modules: (1) the *semantic matching* (SM) module that employs a PLM to perform the semantic matching between questions and relations; (2) the *matching information propagation* (MIP) module that propagates the semantic matching information on KGs. We present the overview of the model architecture in Figure 2. Next, we describe the three parts in detail.

**General Input Formulation.** In order to support both retrieval and reasoning stages, we consider a general form for evaluating entity relevance, where a question $q$ and a subgraph $\mathcal{G}_q$ of candidate entities are given. For the retrieval stage, $\mathcal{G}_q$ is an abstract subgraph that incorporates abstract nodes to merge entities from the same relation. For the reasoning stage, $\mathcal{G}_q$ is constructed based on the

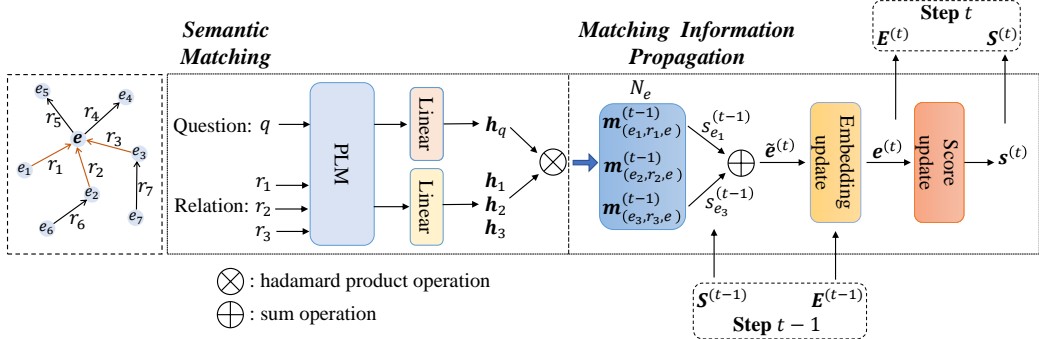

Figure 2: The illustration of updating entity representation $e$ at step $t$ by aggregating the semantic matching information from the set of directed relations pointing to $e$ in the subgraph (*i.e.*, $\{r_1, r_2, r_3\}$) in our UniKGQA.

retrieved subgraph from the retrieval stage, without abstract nodes. Such a general input formulation enables the development of the unified model architecture for the two different stages. In what follows, we will describe the approach in a general way, without considering specific stages.

**Semantic Matching (SM).** The SM module aims to produce the semantic matching features between the question $q$ and a triple $\langle e', r, e \rangle$ from the given subgraph $\mathcal{G}_q$. Considering the excellent modeling capacity of the PLM, we leverage the PLM to produce text encoding as the representations of question $q$ and relation $r$. Specifically, we first utilize the PLM to encode the texts of $q$ and $r$, and employ the output representation of the `[CLS]` token as their representations:

$$\boldsymbol{h}_q = \text{PLM}(q), \ \boldsymbol{h}_r = \text{PLM}(r). \tag{1}$$

Based on $\boldsymbol{h}_q$ and $\boldsymbol{h}_r$, inspired by the NSM model (He et al., 2021), we obtain the vector capturing semantic matching features $\boldsymbol{m}^{(t)}_{\langle e', r, e \rangle}$ between question $q$ and triple $\langle e', r, e \rangle$ at the $t$-th step by adopting corresponding projection layers:

$$\boldsymbol{m}^{(t)}_{\langle e', r, e \rangle} = \sigma \left( \boldsymbol{h}_q \boldsymbol{W}^{(t)}_Q \odot \boldsymbol{h}_r \boldsymbol{W}^{(t)}_R \right), \tag{2}$$

where $\boldsymbol{m}^{(t)}_{\langle e', r, e \rangle} \in \mathbb{R}^d$, $\boldsymbol{W}^{(t)}_Q, \boldsymbol{W}^{(t)}_R \in \mathbb{R}^{h \times d}$ are parameters of the $t$-step projection layers, $h$, $d$ are the hidden dimensions of PLM and the feature vector, respectively, $\sigma$ is the sigmoid activation function, and $\odot$ is the hadamard product.

**Matching Information Propagation (MIP).** Based on the generated semantic matching features, the MIP module first aggregates them to update the entity representation and then utilizes it to obtain the entity match score. To initialize the match score, given a question $q$ and a subgraph $\mathcal{G}_q$, for each entity $e_i \in \mathcal{G}_q$, we set the match score between $q$ and $e_i$ as follows: $s^{(1)}_{e_i} = 1$ if $e_i$ is a topic entity and $s^{(1)}_{e_i} = 0$ otherwise. At the $t$-th step, we utilize the match scores of the head entities computed at the last step $s^{(t-1)}_{e'}$ as the weights and aggregate the matching features from neighboring triples to obtain the representation of the tail entity:

$$\boldsymbol{e}^{(t)} = \boldsymbol{W}^{(t)}_E \left( \left[ \boldsymbol{e}^{(t-1)}; \sum_{\langle e', r, e \rangle \in \mathcal{N}_e} s^{(t-1)}_{e'} \cdot \boldsymbol{m}^{(t)}_{\langle e', r, e \rangle} \right] \right), \tag{3}$$

where $\boldsymbol{e}^{(t)} \in \mathbb{R}^d$ is the representation of the entity $e$ in the $t$-th step, and the $\boldsymbol{W}^{(t)}_E \in \mathbb{R}^{2d \times d}$ is a learnable matrix. At the first step, since there are no matching scores, following the NSM (He et al., 2021) model, we directly aggregate the representations of its one-hop relations as the entity representation: $\boldsymbol{e}^{(1)} = \sigma(\sum_{\langle e', r, e \rangle \in \mathcal{N}_e} \boldsymbol{r} \cdot \boldsymbol{U})$, where the $\boldsymbol{U} \in \mathbb{R}^{2d \times d}$ is a learnable matrix. Based on the representations of all entities $\boldsymbol{E}^{(t)} \in \mathbb{R}^{d \times n}$, we update their entity match scores using the softmax function as:

$$\boldsymbol{s}^{(t)} = \text{softmax} \left( \boldsymbol{E}^{(t)^\top} \boldsymbol{v} \right), \tag{4}$$

where $\boldsymbol{v} \in \mathbb{R}^d$ is a learnable vector.

After $T$-step iterations, we can obtain the final entity match scores $\boldsymbol{s}^{(T)}$, which is a probability distribution over all entities in the subgraph $\mathcal{G}_q$. These match scores can be leveraged to measure the possibilities of the entities being the answers to the given question $q$, and will be used in both the retrieval and reasoning stages.

## 3.2 MODEL TRAINING

In our approach, we have both the retrieval model and the reasoning model for the two stages of multi-hop KGQA. Since the two models adopt the same architecture, we introduce $\Theta$ and $\Gamma$ to denote the model parameters that are used for retrieval and reasoning stages, respectively. As shown in Section 3.1, our architecture contains two groups of parameters, namely the underlying PLM and the other parameters for matching and propagation. Thus, $\Theta$ and $\Gamma$ can be decomposed as: $\Theta = \{\Theta_p, \Theta_o\}$ and $\Gamma = \{\Gamma_p, \Gamma_o\}$, where the subscripts $p$ and $o$ denote the PLM parameters and the other parameters in our architecture, respectively. In order to learn these parameters, we design both pre-training (*i.e.,* question-relation matching) and fine-tuning (*i.e.,* retrieval- and reasoning-oriented learning) strategies based on the unified architecture. Next, we describe the model training approach.

**Pre-training with Question-Relation Matching (QRM).** For pre-training, we mainly focus on learning the parameters of the underlying PLMs (*i.e.,* $\Theta_p$ and $\Gamma_p$). In the implementation, we let the two models share the same copy of PLM parameters, *i.e.,* $\Theta_p = \Gamma_p$. As shown in Section 3.1, the basic capacity of the semantic matching module is to model the relevance between a question and a single relation (Eq. 2), which is based on the text encoding from the underlying PLM. Therefore, we design a contrastive pre-training task based on question-relation matching. Specifically, we adopt the contrastive leaning objective (Hadsell et al., 2006) to align the representations of relevant question-relation pairs while pushing apart others. To collect the relevant question-relation pairs, given an example consisting of a question $q$, the topic entities $\mathcal{T}_q$ and answer entities $\mathcal{A}_q$, we extract all the shortest paths between the $\mathcal{T}_q$ and $\mathcal{A}_q$ from the entire KG and regard all of the relations within these paths as relevant to $q$, denoted as $\mathcal{R}^+$. In this way, we can obtain a number of weak-supervised examples. During pre-training, for each question $q$, we randomly sample a relevant relation $r^+ \in \mathcal{R}^+$, and utilize a contrastive learning loss for pre-training:

$$\mathcal{L}_{PT} = -\log \frac{e^{\mathrm{sim}(\boldsymbol{q}_i, \boldsymbol{r}_i^+)/\tau}}{\sum_{j=1}^{M} \left( e^{\mathrm{sim}(\boldsymbol{q}_i, \boldsymbol{r}_j^+)/\tau} + e^{\mathrm{sim}(\boldsymbol{q}_i, \boldsymbol{r}_j^-)/\tau} \right)} \tag{5}$$

where $\tau$ is a temperature hyperparameter, $r_i^-$ is a randomly sampled negative relation, and $\mathrm{sim}(\boldsymbol{q}, \boldsymbol{r})$ is the cosine similarity, and $\boldsymbol{q}$, $\boldsymbol{r}$ is the question and relation encoded by the PLM from the SM module (Eq. 1). In this way, the question-relation matching capacity will be enhanced by pre-training the PLM parameters. Note that the PLM parameters will be fixed after pre-training.

**Fine-tuning for Retrieval on Abstract Subgraphs (RAS).** After pre-training, we first fine-tune the entire model for learning the parameters $\Theta_o$ according to the retrieval task. Recall that we transform the subgraphs into a form of *abstract subgraphs*, where abstract nodes are incorporated for merging entities from the same relation. Since our MIP module (Section 3.1) can produce the matching scores $\boldsymbol{s}_A$ of nodes in a subgraph (Eq. 4), where the subscript $A$ denotes that the nodes are from an abstract subgraph. Furthermore, we utilize the labeled answers to get the ground-truth vectors, denoted by $\boldsymbol{s}_A^*$. We set an abstract node in $\boldsymbol{s}_A^*$ to 1 if it contains the answer entity. Then we minimize the KL divergence between the learned and ground-truth matching score vectors as:

$$\mathcal{L}_{RAS} = D_{KL}(\boldsymbol{s}_A, \boldsymbol{s}_A^*). \tag{6}$$

After fine-tuning the RAS loss, the retrieval model can be effectively learned. We further utilize it to retrieve the subgraph for the given question $q$, by selecting the top-$K$ ranked nodes according to their match scores. Note that only the node within a reasonable distance to the topic entities will be selected into the subgraph, which ensures a relatively small yet relevant subgraph $\mathcal{G}_q$ for the subsequent reasoning stage to find answer entities.

**Fine-tuning for Reasoning on Retrieved Subgraphs (RRS).** After fine-tuning the retrieval model, we continue to fine-tune the reasoning model by learning the parameters $\Gamma_o$. With the fine-tuned

Table 1: Comparison of different methods.

| Methods | Retrieval | Reasoning | Parameters Transferring |
|---------|-----------|-----------|-------------------------|
| GraftNet | PPR | GraftNet | ✗ |
| PullNet | LSTM | GraftNet | ✗ |
| NSM | PPR | NSM | ✗ |
| SR+NSM | PLM | NSM | ✗ |
| UniKGQA | UniKGQA | UniKGQA | ✓ |

Table 2: Statistics of all datasets.

| Datasets | #Train | #Valid | #Test | Max #hop |
|----------|--------|--------|-------|----------|
| MetaQA-1hop | 96,106 | 9,992 | 9,947 | 1 |
| MetaQA-2hop | 118,980 | 14,872 | 14,872 | 2 |
| MetaQA-3hop | 114,196 | 14,274 | 14,274 | 3 |
| WebQSP | 2,848 | 250 | 1,639 | 2 |
| CWQ | 27,639 | 3,519 | 3,531 | 4 |

retrieval model, we can obtain a smaller subgraph $\mathcal{G}_q$ for each question $q$. In the reasoning stage, we focus on performing accurate reasoning to find the answer entities, so that we recover the original nodes in the abstract nodes and the original relations among them. Since the retrieval and reasoning stages are highly dependent, we first initialize the parameters of the reasoning model with those from the retrieval model: $\Theta_o \to \Gamma_o$. Then, following Eq. 4, we employ a similar approach to fitting the learned matching scores (denoted by $\boldsymbol{s}_R$) with the ground-truth vectors (denoted by $\boldsymbol{s}_R^*$) according to the KL loss:

$$\mathcal{L}_{RRS} = D_{KL}\big(\boldsymbol{s}_R, \boldsymbol{s}_R^*\big), \tag{7}$$

where the subscript $R$ denotes that the nodes come from a retrieved subgraph. After fine-tuning with the RRS loss, we can utilize the learned reasoning model to select the top-$n$ ranked entities as the answer list according to the match scores.

As shown in Figure 1(c), the overall training procedure is composed by: (1) pre-training $\Theta_p$ with question-relation matching, (2) fixing $\Theta_p$ and fine-tuning $\Theta_o$ for retrieval on abstract subgraphs, and (3) fixing the $\Gamma_p$ initialized by $\Theta_p$ and fine-tuning $\Gamma_o$ initialized by $\Theta_o$ for reasoning on subgraphs.

Our work provides a novel unified model for the retrieval and reasoning stages to share the reasoning capacity. In Table 1, we summarize the differences between our method and several popular methods for multi-hop KGQA, including GraphfNet (Sun et al., 2018), PullNet (Sun et al., 2019), NSM (He et al., 2021), and SR+NSM (Zhang et al., 2022). As we can see, existing methods usually adopt different models for the retrieval and reasoning stages, while our approach is more unified. As a major benefit, the information between the two stages can be effectively shared and reused: we initialize the reasoning model with the learned retrieval model.

# 4 EXPERIMENT

## 4.1 EXPERIMENTAL SETTING

**Datasets.** Following existing work on multi-hop KGQA (Sun et al., 2018; 2019; He et al., 2021; Zhang et al., 2022), we adopt three benchmark datasets, namely *MetaQA* (Zhang et al., 2018), *WebQuestionsSP (WebQSP)* (Zhang et al., 2018; Yih et al., 2015), and *Complex WebQuestions 1.1 (CWQ)* (Talmor & Berant, 2018) for evaluating our model. Table 2 shows the statistics of the three datasets. Since previous work has achieved nearly full marks on MetaQA, WebQSP and CWQ are our primarily evaluated datasets. We present a detailed description of these datasets in Appendix A.

**Evaluation Protocol.** For the retrieval performance, we follow Zhang et al. (2022) that evaluate the models by the answer coverage rate (%). It is the proportion of questions whose retrieved subgraphs contain at least one answer. For the reasoning performance, we follow Sun et al. (2018; 2019) that regard the reasoning as a ranking task for evaluation. Given each test question, we rely on the predictive probabilities from the evaluated model to rank all candidate entities and then evaluate whether the top-1 answer is correct with *Hits@1*. Since a question may correspond to multiple answers, we also adopt the widely-used *F1* metric.

**Baselines.** We consider the following baselines for performance comparison: (1) reasoning-focused methods: *KV-Mem* (Miller et al., 2016), *GraftNet* (Sun et al., 2018), *EmbedKGQA* (Saxena et al., 2020), *NSM* (He et al., 2021), *TransferNet* (Shi et al., 2021); (2) retrieval-augmented methods: *PullNet* (Sun et al., 2019), *SR+NSM* (Zhang et al., 2022), *SR+NSM+E2E* (Zhang et al., 2022). We present a detailed description of these baselines in Appendix B.

Table 3: Performance comparison of different methods for KGQA (Hits@1 and F1 in percent). We copy the results for TransferNet from Shi et al. (2021) and others from Zhang et al. (2022). Bold and underline fonts denote the best and the second-best methods, respectively.

| Models | WebQSP | | CWQ | | MetaQA-1 | MetaQA-2 | MetaQA-3 |
|---|---|---|---|---|---|---|---|
| | Hits@1 | F1 | Hits@1 | F1 | Hits@1 | Hits@1 | Hits@1 |
| KV-Mem | 46.7 | 34.5 | 18.4 | 15.7 | 96.2 | 82.7 | 48.9 |
| GraftNet | 66.4 | 60.4 | 36.8 | 32.7 | 97.0 | 94.8 | 77.7 |
| PullNet | 68.1 | - | 45.9 | - | 97.0 | 99.9 | 91.4 |
| EmbedKGQA | 66.6 | - | - | - | 97.5 | 98.8 | 94.8 |
| NSM | 68.7 | 62.8 | 47.6 | 42.4 | 97.1 | 99.9 | 98.9 |
| TransferNet | 71.4 | - | 48.6 | - | 97.5 | **100** | **100** |
| SR+NSM | 68.9 | 64.1 | 50.2 | 47.1 | - | - | - |
| SR+NSM+E2E | 69.5 | 64.1 | 49.3 | 46.3 | - | - | - |
| UniKGQA | 75.1 | 70.2 | 50.7 | 48.0 | 97.5 | 99.0 | 99.1 |
| *w* QU | 77.0 | 71.0 | 50.9 | **49.4** | 97.6 | 99.9 | 99.5 |
| *w* QU,RU | **77.2** | **72.2** | **51.2** | 49.0 | **98.0** | 99.9 | 99.9 |

## 4.2 EVALUATION RESULTS

Table 3 shows the results of different methods on 5 multi-hop KGQA datasets. It can be seen that:

First, most baselines perform very well on the three MetaQA datasets (100% Hits@1). It is because these datasets are based on a few hand-crafted question templates and have only nine relation types for the given KG. Thus, the model can easily capture the relevant semantics between the questions and relations to perform reasoning. To further examine this, we conduct an extra one-shot experiment on MetaQA datasets and present the details in Appendix E. Second, TransferNet performs better than GraftNet, EmbedKGQA, and NSM with the same retrieval method. It attends to question words to compute the scores of relations and transfers entity scores along with the relations. Such a way can effectively capture the question-path matching semantics. Besides, SR+NSM and SR+NSM+E2E outperform NSM and PullNet in a large margin. The reason is that they both leverage a PLM-based relation paths retriever to improve the retrieval performance and then reduce the difficulty of the later reasoning stage.

Finally, on WebQSP and CWQ, our proposed UniKGQA is substantially better than all other competitive baselines. Unlike other baselines that rely on independent models to perform retrieval and reasoning, our approach can utilize a unified architecture to accomplish them. Such a unified architecture can pre-learn the essential capability of question-relation semantic matching for both stages, and is also capable of effectively transferring relevance information from the retrieval stage to the reasoning stage, *i.e.,* initializing the reasoning model with the parameters of the retrieval model.

In our approach, we fix the parameters of the PLM-based encoder for efficiency. Actually, updating its parameters can further improve our performance. Such a way enables researchers to trade off the efficiency and effectiveness when employing our approach in real-world applications. Here, we study it by proposing two variants of our UniKGQA: (1) *w QU* that updates the parameters of the PLM encoder only when encoding questions, (2) *w QU, RU* that updates the parameters of the PLM encoder both when encoding questions and relations. Indeed, both variants can boost the performance of our UniKGQA. And only updating the PLM encoder when encoding questions can obtain a comparable even better performance to update both. A possible reason is that updating the PLM encoder owns when encoding questions and relations may lead to overfitting on the downstream tasks. Therefore, it is promising for our UniKGQA to just update the PLM encoder when encoding questions, as it can achieve better performance with relative less additional computation cost.

## 4.3 FURTHER ANALYSIS

**Retrieval Evaluation.** We evaluate the effectiveness of our UniKGQA to retrieve a smaller but better answer coverage rate subgraph for a given question. Following the evaluation principles of SR (Zhang et al., 2022), we measure such a capacity from three aspects: the direct subgraph size, answer coverage rate, and the final QA performance. Concretely, we first compare UniKGQA with SR (Zhang et al., 2022) and PPR-based heuristic retrieval method (Sun et al., 2018) based on the answer coverage rate curve *w.r.t.* the number of graph nodes. Then, we compare UniKGQA with

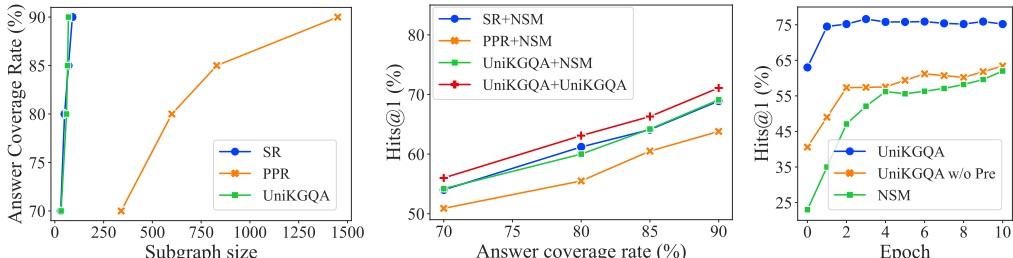

Figure 3: The evaluation of retrieval and fine-tuning efficiency: the answer coverage rate under various subgraph sizes (Left), the Hits@1 scores under various answer coverage rates (Middle), and the Hits@1 scores at different epochs on WebQSP (Right).

SR+NSM (Zhang et al., 2022) and PPR+NSM (He et al., 2021) based on their final QA performance. To further study the effectiveness of our approach, we add an extra variant of our UniKGQA, namely UniKGQA+NSM, which relies on UniKGQA for retrieval while NSM for performing reasoning. The left and middle of Figure 3 show the comparison results of the above methods. As we can see, under the same size of retrieved subgraphs, UniKGQA and SR have significantly larger answer coverage rates than PPR. It demonstrates the effectiveness and necessity of training a learnable retrieval model. Besides, although the curves of UniKGQA and SR are very similar, our UniKGQA can achieve a better final reasoning performance than SR+NSM. The reason is that UniKGQA can transfer the relevance information from the retrieval stage to the reasoning stage based on the unified architecture, learning a more effective reasoning model. Such a finding can be further verified by comparing our UniKGQA with UniKGQA+NSM.

Table 4: Ablation study of our training strategies.

| Models | WebQSP | | CWQ | |
|---|---|---|---|---|
| | Hits@1 | F1 | Hits@1 | F1 |
| UniKGQA w QU | 77.0 | 71.0 | 50.9 | 49.4 |
| *w/o* Pre | 75.4 | 70.6 | 49.2 | 48.8 |
| *w/o* Trans | 75.8 | 70.6 | 49.8 | 49.3 |
| *w/o* Pre, Trans | 72.5 | 60.0 | 48.1 | 48.4 |

**Ablation Study.** Our UniKGQA contains two important training strategies to improve performance: (1) pre-training with question-relation matching, (2) initializing the parameters of the reasoning model with the retrieval model. Here, we conduct the ablation study to verify their effectiveness. We propose three variants as: (1) *w/o Pre* removing the pre-training procedure, (2) *w/o Trans* removing the initialization with the parameters of retrieval model, (3) *w/o Pre, Trans* removing both the pre-training and initialization procedures. We show the results of the ablation study in Table 4. We can see that all these variants underperform the complete UniKGQA, which indicates that the two training strategies are both important for the final performance. Besides, such an observation also verifies that our UniKGQA is indeed capable of transferring and reusing the learned knowledge to improve the final performance.

**Fine-tuning Efficiency.** As our UniKGQA model can transfer the learned knowledge from the pre-training stage and the retrieval task, it can be easily adapted into downstream reasoning tasks. In this way, we can perform a more efficient fine-tuning on the reasoning task with a few fine-tuning steps. To explore it, we compare the performance changes of our UniKGQA with a strong baseline NSM *w.r.t.* the increasing of fine-tuning epochs based on the same retrieved subgraphs. The results are presented on the right of Figure 3. First, we can see that before fine-tuning (*i.e.,* when the epoch is zero), our UniKGQA has achieved a comparable performance as the best results of NSM at the last epoch. It indicates that the reasoning model has successfully leveraged the knowledge from prior tasks based on the parameters initialized by the retrieval model. After fine-tuning with two epochs, our UniKGQA has already achieved a good performance. It verifies that our model can be fine-tuned in an efficient way with very few epochs. To further investigate our UniKGQA model, we conduct parameter sensitivity analysis *w.r.t.* pre-training steps, hidden dimensions, and the number of retrieved nodes $K$, shown in Appendix H.

## 5 RELATED WORK

**Multi-hop Knowledge Graph Question Answering.** Multi-hop KGQA aims to seek answer entities that are multiple hops away from the topic entities in a large-scale KG. Considering the efficiency and accuracy, existing work (Sun et al., 2018; 2019; Zhang et al., 2022) typically first retrieves a question-relevant subgraph to reduce the search space and then performs multi-hop reasoning on it. Such a retrieval-and-reasoning paradigm has shown superiority over directly reasoning on the entire KG (Chen et al., 2019; Saxena et al., 2020).

The retrieval stage focuses on extracting a relatively small subgraph involving the answer entities. A commonly-used approach is to collect entities with nearer hops around the topic entities to compose the subgraph and filter the ones with low Personalized PageRank scores to reduce the graph size (Sun et al., 2018; He et al., 2021). Despite the simplicity, such a way neglects the question semantics, limiting the retrieval efficiency and accuracy. To address it, several work (Sun et al., 2019; Zhang et al., 2022) devises retrievers based on semantic matching using neural networks (*e.g.,* LSTMs or PLMs). Starting from the topic entities, these retrievers iteratively measure the semantic relevance between the question and neighbouring entities or relations, and add proper ones into the subgraph. In this way, a smaller but more question-relevant subgraph would be constructed.

The reasoning stage aims to accurately find the answer entities of the given question by walking along the relations starting from the topic entities. Early work (Miller et al., 2016; Sun et al., 2018; 2019; Jiang et al., 2022) relies on the special network architectures (*e.g.,* Key-Value Memory Network or Graph Convolution Network) to model the multi-hop reasoning process. Recent work further enhances the reasoning capacity of the above networks from the perspectives of intermediate supervision signals (He et al., 2021), knowledge transferring (Shi et al., 2021), etc. However, all these methods design different model architectures and training methods for the retrieval and reasoning stages, respectively, neglecting the similarity and intrinsic connection of the two stages.

Recently, some work parses the question into structured query language (*e.g.,* SPARQL) (Lan et al., 2021; Das et al., 2021; Huang et al., 2021) and executes it by a query engine to get answers. In this way, the encoder-decoder architecture (*i.e.,* T5 (Raffel et al., 2020)) is generally adopted to produce the structured queries, where the annotated structured queries are also required for training.

**Dense Retrieval.** Given a query, the dense retrieval task aims to select relevant documents from a large-scale document pool. Different from the traditional sparse term-based retrieval methods, *e.g.,* TF-IDF (Chen et al., 2017) and BM25 (Robertson & Zaragoza, 2009), dense retrieval methods (Karpukhin et al., 2020; Zhou et al., 2022a;b) rely on a bi-encoder architecture to map queries and documents into low-dimensional dense vectors. Then their relevance scores can be measured using vector distance metrics (*e.g.,* cosine similarity), which supports efficient approximate nearest neighbour (ANN) search algorithms. In multi-hop KGQA, starting from the topic entities, we need to select the relevant neighboring triples from a large-scale KG, to induce a path to reach the answer entities, which can be seen as a constrained dense retrieval task. Therefore, in this work, we also incorporate a bi-encoder architecture to map questions and relations into dense vectors, and then perform retrieval or reasoning based on their vector distances.

## 6 CONCLUSION

In this work, we proposed a novel approach for the multi-hop KGQA task. As the major technical contribution, UniKGQA introduced a unified model architecture based on PLMs for both retrieval and reasoning stages, consisting of the semantic matching module and the matching information propagation module. To cope with the different scales of search space in the two stages, we proposed to generate abstract subgraphs for the retrieval stage, which can significantly reduce the number of nodes to be searched. Furthermore, we designed an effective model learning method with both pre-training (*i.e.,* question-relation matching) and fine-tuning (*i.e.,* retrieval- and reasoning-oriented learning) strategies based on the unified architecture. With the unified architecture, the proposed learning method can effectively enhance the sharing and transferring of relevance information between the two stages. We conducted extensive experiments on three benchmark datasets, and the experimental results show that our proposed unified model outperforms the competitive methods, especially on more challenging datasets (*i.e.,* WebQSP and CWQ).

ACKNOWLEDGMENTS

This work was partially supported by National Natural Science Foundation of China under Grant No. 62222215, Beijing Natural Science Foundation under Grant No. 4222027, and Beijing Outstanding Young Scientist Program under Grant No. BJJWZYJH012019100020098. And this work is also partially supported by the Outstanding Innovative Talents Cultivation Funded Programs 2022 of Renmin University of China. Xin Zhao is the corresponding author.

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

Table 5: Analysis of MetaQA datasets.

| Dataset | # of templates | average # of training cases per template | # of used relations for question construction |
|---|---|---|---|
| MetaQA-1 | 161 | 597 | 9 |
| MetaQA-2 | 210 | 567 | 9 |
| MetaQA-3 | 150 | 761 | 9 |

# A  DATASETS

We adopt three widely-used multi-hop KGQA datasets in this work:

• **MetaQA** (Zhang et al., 2018) contains more than 400k questions in the domain of movie and the answer entities are up to 3 hops away from the topic entities. According to the number of hops, this dataset is split into three sub-datasets, *i.e.,* MetaQA-1hop, MetaQA-2hop, and MetaQA-3hop.

• **WebQuestionsSP (WebQSP)** (Yih et al., 2015) contains 4,737 questions and the answer entities require up to 2-hop reasoning on the KG Freebase (Bollacker et al., 2008). We use the same train/valid/test splits as GraftNet (Sun et al., 2018).

• **Complex WebQuestions 1.1 (CWQ)** (Talmor & Berant, 2018) is constructed based on WebQSP by extending the question entities or adding constraints to answers. These questions require up to 4-hop reasoning on the KG Freebase (Bollacker et al., 2008).

Existing work has demonstrated that the training data for MetaQA is *more than sufficient* (Shi et al., 2021; He et al., 2021), hence all the comparison methods in our experiments can achieve very high performance. We conduct further analysis of the three MetaQA datasets about the number of templates, the average number of training cases per template, and the number of relations used for constructing questions, and show them in Table 5. In summary, more training cases and simpler questions make the MetaQA easier to be solved.

# B  BASELINES

We consider the following baseline methods for performance comparison:

• **KV-Mem** (Miller et al., 2016) maintains a key-value memory table to store KG facts, and conducts multi-hop reasoning by performing iterative read operations on the memory.

• **GraftNet** (Sun et al., 2018) first retrieves the question-relevant subgraph and text sentences from the KG and Wikipedia respectively with a heuristic method. Then it adopts a graph neural network to perform multi-hop reasoning on a heterogeneous graph built upon the subgraph and text sentences.

• **PullNet** (Sun et al., 2019) trains a graph retrieval model composed of a LSTM and a graph neural network instead of the heuristic way in GraftNet for the retrieval task, and then conducts multi-hop reasoning with GraftNet.

• **EmbedKGQA** (Saxena et al., 2020) reformulates the multi-hop reasoning of GraftNet as a link prediction task by matching pre-trained entity embeddings with question representations from a PLM.

• **NSM** (He et al., 2021) first conducts retrieval following GraftNet and then adapt the neural state machine (Hudson & Manning, 2019) used in visual reasoning for multi-hop reasoning on the KG.

• **TransferNet** (Shi et al., 2021) first conducts retrieval following GraftNet and then performs the multi-hop reasoning on a KG or a text-formed relation graph in a transparent framework. The reasoning model consists of a PLM for question encoding and a graph neural network for updating the relevance scores between entities and the question.

• **SR+NSM** (Zhang et al., 2022) first learns a PLM-based relation path retriever to conduct effectively retrieval and then leverages NSM reasoner to perform multi-hop reasoning.

• **SR+NSM+E2E** (Zhang et al., 2022) further fine-tunes the SR+NSM by an end-to-end way.

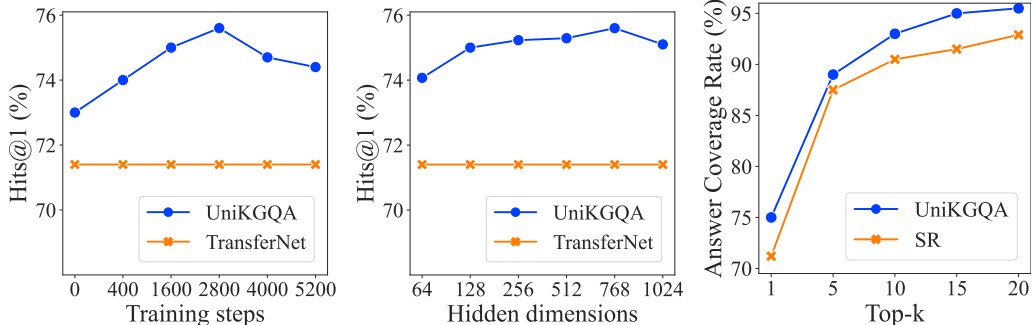

Figure 4: The results of ablation study on WebQSP. The performance on WebQSP of varying pre-training steps (Left), hidden dimensions (Middle), and the number of retrieved nodes $K$ (Right).

## C   KNOWLEDGE GRAPH PREPROCESSING DETAILS

We preprocess the full Freebase following existing work (Sun et al., 2018; He et al., 2021). For MetaQA, we directly use the subset of WikiMovies provided by the datasets, and the size is about 134,741. For WebQSP and CWQ datasets, we set the max hops of retrieval and reasoning as two and four, respectively. Based on the topic entities labeled in original datasets, we reserve the neighbor-hood subgraph consisting of entities within the four hops of the topic entities for each sample. After such simple preprocessing, the size of KG we used is 147,748,092 for WebQSP and 202,358,414 for CWQ. Based on the preprocessed KG, we conduct the retrieval and reasoning using our proposed approach.

## D   IMPLEMENTATION DETAILS.

During pre-training, we collect question-relation pairs based on the shortest relation paths between topic entities and answer entities, and then use these pairs to pre-train the RoBERTa-base (Liu et al., 2019) model with the contrastive learning objective. We set the temperature $\tau$ as 0.05, and select the best model by evaluating *Hits@1* on the validation set. For retrieval and reasoning, we initialize the PLM module of our UniKGQA model with the contrastive learning pre-trained RoBERTa, and set the hidden size of other linear layers as 768. We optimize parameters with the AdamW optimizer, where the learning rate is 0.00001 for the PLM module, and 0.0005 for other parameters. The batch size is set to 40. The reasoning step is set to 4 for CWQ dataset, 3 for WebQSP and MetaQA-3 datasets, 2 for MetaQA-2 dataset, and 1 for MetaQA-1 dataset. We preprocess the KGs for each datasets following existing work (Sun et al., 2018; He et al., 2021).

## E   ONE-SHOT EXPERIMENT FOR METAQA

Table 6: One-shot experiment results on MetaQA (Hits@1 in percent).

| Model | MetaQA-1 | MetaQA-2 | MetaQA-3 |
|---|---|---|---|
| NSM | 94.8 | 97.0 | 91.0 |
| TransferNet | 96.5 | 97.5 | 90.1 |
| UniKGQA | 97.1 | 98.2 | 92.6 |

Since the samples in MetaQA are more than sufficient, all the comparison methods in our experiments have achieved very high performance. For example, our method and previous work (*e.g.,* TransferNet and NSM) have achieved more than 98% Hits@1 on MetaQA, which shows that this dataset's performance may have been saturated. To examine this assumption, we consider conducting few-shot experiments to verify the performance of different methods. Specially, we follow the NSM paper (He et al., 2021) that conducts the one-shot experiment. We randomly sample just one training case for each question template from the original training set, to form a one-shot training dataset. In this way, the numbers of training samples for MetaQA-1, MetaQA-2, and MetaQA-3 are 161, 210, and 150, respectively. We evaluate the performance of our approach and some strong

baselines (*i.e.,* TrasnferNet and NSM) trained with this new training dataset. As shown in Table 6, our method can consistently outperform these baselines in all three subsets.

# F    ABLATION STUDY OF OUR UNIFIED MODEL ARCHITECTURE

Table 7: Ablation study by combining our UniKGQA with other models.

| Models | WebQSP (Hits@1) | CWQ (Hits@1) |
|---|---|---|
| PPR+NSM | 68.7 | 47.6 |
| SR+NSM | 68.9 | 50.2 |
| SR+UniKGQA | 70.5 | 48.0 |
| UniKGQA+NSM | 69.1 | 49.2 |
| UniKGQA+UniKGQA | 75.1 | 50.7 |

The unified model architecture is the key of our approach. Once the unified model architecture is removed, it would be hard to share the question-relation matching capability enhanced by pre-training in retrieval and reasoning stages, and also hard to transfer the relevance information for multi-hop KGQA learned in the retrieval stage to the reasoning stage. To verify it, we conduct an extra ablation study to explore the effect of only adopting the unified model architecture as the reasoning model or the retrieval model. We select the existing strong retrieval model (*i.e.,* SR) and reasoning model (*i.e.,* NSM), and compare the performance when integrated with our UniKGQA. As we can see in Table 7, all the variants underperform our UniKGQA. It indicates that the unified model used in the retrieval and reasoning stages simultaneously is indeed the key reason for improvement.

# G    ANALYSIS OF THE PRE-TRAINING STRATEGY

Table 8: Results of variants with or without pre-training strategy (Pre) and updating the PLM (QU).

| Models | WebQSP (Hits@1) | CWQ (Hits@1) |
|---|---|---|
| UniKGQA | 75.1 | 50.7 |
| w QU | 77.0 | 50.9 |
| w/o Pre, w QU | 75.4 | 49.2 |
| w/o Pre | 67.3 | 48.1 |

We conduct the analysis experiments to investigate how the pre-training strategy (Pre) affects the performance with or without updating the PLM (QU). We show the results in Table 8. Once removing the pre-training strategy, the model performance would drop 10.4% (2.1%) in WebQSP and 5.1% (3.3%) in CWQ when fixing (not fixing) the PLM. It indicates that the pre-training strategy is an important component of our approach. After pre-training, the PLM can be fixed for more efficient parameters optimization during fine-tuning.

# H    PARAMETER SENSITIVITY ANALYSIS

**Pre-training Steps** Although the pre-training strategy has shown effective in our approach, too many pre-training steps will be time-consuming and costly. Here, we investigate the performance with respect to varying pre-training steps. As shown in the left of Figure 4, we can see that our method can reach the best performance with only few pre-training steps (*i.e.,* 2800) compared with the best baseline TransferNet. It shows that our approach does not require too many steps for pre-training. Instead, we can see that too many pre-training steps will hurt the model performance. The reason may be that the PLM has overfit into the contrastive learning objective.

**Parameter Tuning.** In our approach, we have two hyper-parameters required to tune: (1) the hidden size of linear layers $d$ and (2) the number of retrieved nodes $K$. Here, we tune the $d$ amongst $\{64, 128, 256, 512, 768, 1024\}$ and $K$ amongst $\{1, 5, 10, 15, 20\}$. We show the results in the middle and right of Figure 4 compared with the best results for the reasoning stage and the retrieval stage. Since $K$ is a consistent hyper-parameter in the UniKGQA and SR, we also describe the various results of SR with different $K$ to give a fair comparison. First, we can see that our method is robust to

different hidden sizes, as the performance is consistently nearby 77.0. As the PLM adopts 768 as the embedding size, we can see 768 is also slightly better than other numbers. Besides, we can see that with the increase of $K$, the answer coverage rate also improves consistently. However, when $K$ increases to 15 or even 20, the performance gain becomes relatively small. It means that the retrieved subgraphs are likely saturated, and further increasing $K$ could only bring marginal improvement.

