# OpenReview forum: "UniKGQA: Unified Retrieval and Reasoning for Solving Multi-hop Question Answering Over Knowledge Graph"
_ICLR.cc/2023/Conference — ICLR 2023 poster_

### Official Review · Reviewer_Xn2s · 2022-10-20

**Confidence:** 3
**Correctness:** 3
**Technical Novelty And Significance:** 3
**Empirical Novelty And Significance:** 3
**Recommendation:** 6

**Clarity, Quality, Novelty And Reproducibility:**

Clarity can be improved. The writing could be improved. Figures are too complicated.

Quality of the work is ok: the paper did extensive experiments on different dataset to help understand the performance deltas.

Novelty is less significant: Using the sample pre-trained model feels more incremental and the benefit is only justified by better evaluation results.

Reproducibility is not clear. The authors didn't mention open-source or other means to reproduce.

**Strength And Weaknesses:**

Strength

1) Evaluation results: The evaluation results on WebQSP showed clear advantages of the proposed model. Ablation study and comparison under different setup helped support the paper's claim.

Weakness

1) Writing can be improved. Key abbreviations are not explained at first use (PLM / PPR / topic entity etc, it's always good to reduce reader's guess work). There are typos and proof-reading misses.

2) The explanation of Abstract Subgraph is not clear. What if the tail entities in the same set branch out to different 2nd-hop entities? It's not all clear to the Reviewer. An example would help.

3) It's better to explain more on the key differences between the WebQSP/CWQ sets and MetaQA, as UniKGQA shows clear improvements on the former but not the latter. Giving some examples would help too.

List of concrete issues that should be improved.
1) Figure 2. the order of h_m and h_q seems reversed.
2) Page 4, MIP section, it's better to explain "topic entity" when used first time.
3) Page 4 to the bottom, "are a learnable vector" -> "is a ..."; "of if the entities" -> "of the entities being"?
4) Page 5 to the bottom, "Eq 4" should be "Eq 6"?
5) P7 Section 4.2 second paragraph, it's not clear if unified model is the key reason of improvement, we need more supporting evidence.
6) P8 Table 4, Why use "Trans"?

**Summary Of The Paper:**

The paper proposed a model to do multi-hop question-answering over Knowledge Graphs. The core model responsible for relevance is trained once and shared between the initial retrieval of the subgraph and the detailed reasoning phases. A key claim of the paper is that by unifying the model for the two phases, we can get better quality in the answers than doing them separately.

The proposed UniKGQA model consists of two main parts: Pretrained Language Model (PLM)-based semantic matching and the propagation of matching information through the graphs. The PLM (RoBERTa in implementation) is fine-tuned with task specific data to capture the connection between entities and relationships. Its parameters are shared between the retrieval and reasoning phases.

Evaluation was done on three different datasets. The UniKGQA model was compared with recent baselines and showed on-par results on two of them and significantly better results on WebQSP. The paper included detailed ablation study and different fine-tuning setups to show the benefit of using a joint model for two phrases.


**Summary Of The Review:**

The Reviewer suggest weak accept of the paper after fixing the writing issues.

The paper's main contribution is clear and justified by the evaluation result, that using a unified model does show advantages in overall and break-down comparison with other recent methods.

---

> ### Author Response · Authors · 2022-11-19
> **Response to the Concerns of the Reviewer [Part 2]**
>
> **4.List of concrete issues that should be improved.**
>
> Figure 2. the order of h_m and h_q seems reversed.
>
> Page 4, MIP section, it's better to explain "topic entity" when used first time.
>
> Page 4 to the bottom, "are a learnable vector" -> "is a ..."; "of if the entities" -> "of the entities being"?
>
> Page 5 to the bottom, "Eq 4" should be "Eq 6"?
>
> P7 Section 4.2 second paragraph, it's not clear if unified model is the key reason of improvement, we need more supporting evidence.
>
> P8 Table 4, Why use "Trans"?
>
> - For issues (1) and (3), we modify these typos in our new manuscript.
> - For issue (2), we have explained the "topic entity" in the second paragraph of the preliminary section (Page 3 to the middle).
> - For issue (4), since we unify the model architecture and share a general input formulation for the retrieval and reasoning stages, both the reasoning and the retrieval models can rely on the predicted entities matching scores with Eq 4 to evaluate the question-entity relevance. Then, the reasoning and retrieval models update parameters with their specific ground-truth labels, using Eq 6 and Eq7, respectively.
> - For issue (5), the unified model architecture is the key of our approach. Once the unified model architecture is removed, it is hard to share the question-relation matching capability enhanced by pre-training in retrieval and reasoning stages, and also hard to transfer the relevance information for multi-hop KGQA learned in the retrieval stage to the reasoning stage. To verify it, we conduct an extra ablation study to explore the effect of only adopting the unified model architecture as the reasoning model or the retrieval model. We select the existing strong retrieval model (i.e., SR) and reasoning model (i.e., NSM) and compare the performance when used with our UniKGQA. As we can see, all the variants underperform the UniKGQ, which indicates that the unified model used in the retrieval and reasoning stages simultaneously is indeed the key reason for improvement.
>
> \\begin{array} {|l|c|c|}
> \\hline
> & \\textrm{WebQSP (Hits@1)} & \\textrm{CWQ (Hits@1)} \\\\
> \\hline
> \\textrm{PPR+NSM} & 68.7 & 47.6 \\\\
> \\hline
> \\textrm{SR+NSM} & 68.9 & 50.2 \\\\
> \\hline
> \\textrm{SR+UniKGQA}  & 70.5 & 48.0 \\\\
> \\hline
> \\textrm{UniKGQA+NSM} & 69.1 & 49.2 \\\\
> \\hline
> \\textrm{UniKGQA+UniKGQA}  & 75.1 & 50.7 \\\\
> \\hline
> \\end{array}
>
> - For issue (6), "Trans" means transferring the relevance information from the retrieval stage to the reasoning stage by initializing the reasoning model with the learned retrieval model.
>
> **5. Novelty is less significant: Using the sample pre-trained model feels more incremental and the benefit is only justified by better evaluation results.**
>
> Although the idea is intuitive, a simple PLM is hard to work well on the KBQA task. The reasons are threefold.
> - (1) First, the two stages have different input and output settings.
> - (2) Second, it is not clear how to transfer useful information across the two stages except for sharing the parameters.
> - (3) Third, it is unclear how to train such a PLM-based approach, which involves multi-stage pre-training and fine-tuning.
>
> To our knowledge, these issues are seldom considered in previous work, and we don't find any related solutions in the literature of previous KBQA solutions. Our approach is the first work that unifies the retrieval and reasoning in both model architecture and learning for the multi-hop KGQA task. The specific technical contributions are (according to the above issues):
> - (1) We introduce the concept of abstract graphs, so as to unify the input for both stages (since the retrieval stage faces a significantly larger scale of input).
> - (2) We divide the parameters into two major parts: underlying PLM (group 1) and other parameters (group 2). We share the parameters in group 1 and utilize the parameters of group 1 from the first stage to initialize those for the second stage.
> - (3) We propose to first pre-train the underlying PLM with question-relation semantic matching and fix it when fine-tuning on the two stages. After pre-training, we first fine-tune the retrieval model on abstract subgraphs. Then, we consider utilizing the learned retrieval model to initialize the reasoning model and continue to fine-tune the reasoning model on retrieved subgraphs.
> As discussed above, we can see that it is non-trivial to develop a PLM-based approach for large-scale KBQA in a multi-stage procedure. Our work makes the first attempt and achieves very promising results.
>
> **6. Reproducibility is not clear. The authors didn't mention open-source or other means to reproduce.**
>
> We upload our code into the supplementary material and will release all the code, data and checkpoint for the reproduction of all the experiments after the review.

---

> ### Author Response · Authors · 2022-11-19
> **Response to the Concerns of the Reviwers [Part 1]**
>
> Thanks for your insightful suggestions and we have listed our response to your concerns as follows. If you also have any other questions, please feel free to let us know. We will continue to try our best to answer for you.
>
> **1.Writing can be improved. Key abbreviations are not explained at first use (PLM / PPR / topic entity etc, it's always good to reduce reader's guess work). There are typos and proof-reading misses.**
>
> Thanks for your careful reading. We have checked and added the explanation of the key abbreviations and carefully revised the typos and proofreading misses.
>
> **2.The explanation of Abstract Subgraph is not clear. What if the tail entities in the same set branch out to different 2nd-hop entities? It's not all clear to the Reviewer. An example would help.**
>
> We have modified the illustrative example in Figure 1 to show a more intuitive transformation from the subgraph (Figure 1(a)) to the abstract subgraph (Figure 1(b)). We construct the abstract subgraph from the topic entities to their neighborhood nodes and then broadcast it to farther nodes. For example, given an abstract node (e.g., [a,b]) consisting of two entities (e.g., a, b), we first select its next-hop triples from the KG (e.g., a--r1--c, b--r1--e, a--r2--d), and then merge the tail entities with the same relation into a new abstract node (e.g., [c,e], [d]). As a result, we can utilize just two triples to represent the above triples (e.g., [a,b]--r1--[c,e], [a,b]--r2--[d] ).
> Note that the KG often contains a large amount of one-to-multiple formatted factual triples [1] (i.e., a head entity can derive multiple tail entities through a relation). This characteristic of KG is the basis for reducing the graph size effectively in the Abstract Subgraph.
>
> [1] Kurt D. Bollacker, Colin Evans, Praveen K. Paritosh, Tim Sturge, and Jamie Taylor. Freebase: a collaboratively created graph database for structuring human knowledge. SIGMOD, 2008.
>
> **3.It's better to explain more on the key differences between the WebQSP/CWQ sets and MetaQA, as UniKGQA shows clear improvements on the former but not the latter. Giving some examples would help too.**
>
> The major difference between these datasets lies in the sample sufficiency w.r.t. to the number of relations or the number of templates. Intuitively, the more relations and the more templates (since they are constructed based on templates to cover different relations), the more samples are required to train the KBQA models.
>
> As NSM [1] mentioned, the training data for MetaQA is ***more than sufficient***. We conduct further analysis of the three datasets about the number of templates, the average number of training cases per template, and the number of relations used for constructing questions. In summary, more training cases and simpler questions make the MetaQA easier to be solved.
>
> \\begin{array} {|l|c|c|}
> \\hline
> & \\textrm{number of templates} & \\textrm{average number of training cases per template} & \\textrm{number of used relations in question} \\\\
> \\hline
> \\textrm{MetaQA-1} & 161 & 597 & 9 \\\\
> \\hline
> \\textrm{MetaQA-2}  & 210 & 567 & 9 \\\\
> \\hline
> \\textrm{MetaQA-3} & 150 & 761 & 9 \\\\
> \\hline
> \\end{array}
>
>
>
>
> Since the samples in MetaQA are sufficient, all the comparison methods in our experiments have achieved very high performance. For example, our method and previous work (e.g., TransferNet, NSM) have achieved more than 98% on MetaQA, which shows that this dataset's performance may have been saturated.
>
> To examine this assumption, we consider conducting few-shot experiments to verify the performance of different methods. Specially, we follow the NSM paper that conducts a one-shot experiment. We randomly sample just one training case for each question template from the original training set, to form a one-shot training dataset. The number of training samples for MetaQA-1, MetaQA-2, and MetaQA-3 are 161, 210, and 150, respectively. We evaluate the performance of our approach and some strong baselines (i.e., TrasnferNet and NSM) trained with this new training dataset. As shown in the following Table, our method can consistently outperform these baselines in all three subsets.
>
> \\begin{array} {|l|c|c|}
> \\hline
> & \\textrm{MetaQA-1} & \\textrm{MetaQA-2} & \\textrm{MetaQA-3} \\\\
> \\hline
> \\textrm{NSM} & 94.8 & 97.0 & 91.0 \\\\
> \\hline
> \\textrm{TransferNet}  & 96.5 & 97.5 & 89.5 \\\\
> \\hline
> \\textrm{UniKGQA} & 97.1 & 98.2 & 92.6 \\\\
> \\hline
> \\end{array}
>
> [1] Gaole He, Yunshi Lan, Jing Jiang, Wayne Xin Zhao, and Ji-Rong Wen. WSDM, 2021.

---

> ### Author Response · Authors · 2022-12-06
> **Kindly Reminder for the Discussion**
>
> Dear Reviewer Xn2s,
>
> Thanks for your careful reading of our paper. We have tried our best to elaborate the unclear points (e.g., Abstract Subgraph) and revised our paper accordingly. We have added the one-shot experiments on MetaQA in Appendix E.  Besides, we have uploaded the code of our approach in the supplementary material for reproducibility. We would like to know whether you find our response satisfactory, or if there are more questions that we could clarify. Since the rebuttal stage is coming to an end, we are more than happy to hear your comments and address any of your further concerns during the remaining time.
>
> Best,
>
> Authors

---

> > ### Comment · Reviewer_Xn2s · 2022-12-09
> > **The authors' response resolved most of my concern.**
> >
> > Thank you for the detailed responses to my questions. My concerns have been resolved with the answers, esp on clarity,  imbalance performance on different datasets and reproducibility. I update my recommendation to accept.

---

> > > ### Author Response · Authors · 2022-12-10
> > > **Appreciate your new response and updated score**
> > >
> > > Dear Reviewer Xn2s,
> > >
> > > **We sincerely thank you for the positive reply and the continued vote to accept the work!**
> > >
> > > We have prepared our code into an anonymous GitHub Webpage, i.e.,https://anonymous.4open.science/r/UniKGQA-C985/README.md, and show how to pre-train and fine-tune the unified model (UniKGQA) using our provided code. After the reviewer period, we will further make the trained checkpoints public for the utilization of other researchers. We sincerely appreciate your constructive suggestions!
> > >
> > > Best,
> > > Authors

---

> > > ### Author Response · Authors · 2022-12-11
> > > **A friendly reminder**
> > >
> > > Dear Reviewer Xn2s,
> > >
> > > We thank you once more for your positive comment. We would like to friendly remind you that the recommendation score in the official review is not changed yet. And we sincerely hope that you consider updating the score if it is inconsistent with your reply comments.
> > >
> > > We are very grateful for your support of our paper and your time and effort in reviewing our paper.
> > >
> > > Best,
> > >
> > > Authors

---

### Official Review · Reviewer_Zm6y · 2022-10-24

**Confidence:** 3
**Clarity, Quality, Novelty And Reproducibility:** This paper is well written and the so…
**Correctness:** 3
**Technical Novelty And Significance:** 3
**Empirical Novelty And Significance:** 2
**Recommendation:** 8

**Strength And Weaknesses:**

Strength.
This paper proposes a novel solution which uses pretrained language models to fuse the retrieval and reasoning stages for knowledge graph question answering.

Weakness.
1.	Please present more implementation details about the baselines.

2.	It seems that most of the baselines are GNN-based models, which do not include additional knowledge. Whereas the solution in this paper uses pretrained models to introduce additional knowledge, which makes the experiments less comparable and persuasive.

3.	To render this paper more convincing, I suggest the authors present more baseline models which use similar pretrained models to do this task. To name a few:
a.	Xin Huang, Jung-Jae Kim, and Bowei Zou. 2021. Unseen Entity Handling in Complex Question Answering over Knowledge Base via Language Generation. In Findings of the Association for Computational Linguistics: EMNLP 2021, pages 547–557, Punta Cana, Dominican Republic. Association for Computational Linguistics.

b.	Rajarshi Das, Manzil Zaheer, Dung Thai, Ameya Godbole, Ethan Perez, Jay Yoon Lee, Lizhen Tan, Lazaros Polymenakos, and Andrew McCallum. 2021. Case-based Reasoning for Natural Language Queries over Knowledge Bases. In Proceedings of the 2021 Conference on Empirical Methods in Natural Language Processing, pages 9594–9611, Online and Punta Cana, Dominican Republic. Association for Computational Linguistics.

**Summary Of The Paper:**

This paper proposes a novel solution which uses pretrained language models to fuse the retrieval and reasoning stages for knowledge graph question answering. The authors also conduct extensive experiments to verify the effectiveness of the proposed model.

**Summary Of The Review:**

This paper proposes a novel solution which uses pretrained language models to fuse the retrieval and reasoning stages for knowledge graph question answering. The paper is well written and the solution is somewhat novelty. However, the experiments lack comparison with some latest work.

---

> ### Author Response · Authors · 2022-11-19
> **Response to the Concerns of Reviewer [Part 2]**
>
> **3.This paper proposes a novel solution which uses pretrained language models to fuse the retrieval and reasoning stages for knowledge graph question answering. The paper is well written and the solution is somewhat novelty. However, the experiments lack comparison with some latest work.**
>
> We believe this query is related to query 2. Please refer to the above response about the baseline selection in our approach.
>
> As shown in the above table, under the same problem formulation where only the answer entities are given as labels, we have compared our method with relatively comprehensive baselines, including many competitive and latest PLM-enhanced approaches.
> For example, TransferNet, SR+NSM, and SR+NSM+E2E were published in 2021, 2022, and 2022. As the results are shown in Table 3 of our manuscript, our approach mostly outperforms all other baselines, indicating our proposed approach's effectiveness.
>
> Although there is some recent work, as the reviewer suggested, they are not focused on the same task setting as our approach. Incorporating them into performance will lead to an unfair comparison.

---

> ### Author Response · Authors · 2022-11-19
> **Response to the Concerns of Reviewer [Part 1]**
>
> Thanks for your insightful suggestions and we have listed our response to the concerns about the experiment fairness as follows. If you still have any other questions, please do not hesitate to tell us. We will continue to try our best to answer for you.
>
> **1.Please present more implementation details about the baselines.**
>
> Due to the page limitation, we just show the major difference between the baselines and our approach from the perspectives of how they implement the retrieval and reasoning modules. Following the reviewer's suggestion, we give a more detailed description of all baselines in the Appendix B. These baselines are implemented and tuned, and we will also consider releasing our implementations of these baselines.
>
> **2.It seems that most of the baselines are GNN-based models, which do not include additional knowledge. Whereas the solution in this paper uses pretrained models to introduce additional knowledge, which makes the experiments less comparable and persuasive. To render this paper more convincing, I suggest the authors present more baseline models which use similar pretrained models to do this task.**
>
> Currently, we have eight baseline methods for comparison in total. Actually, there are four baselines that also adopt the pre-trained language model:  EmbedKGQA, TransferNet, SR+NSM, and SR+NSM+E2E. As responded in query 1, we have provided all the necessary details for implementing these baselines, and highlighted what has used PLMs in implementation. These baselines are widely used in KBQA, comprehensively covering the necessary comparisons in the literature.
>
> We thank the reviewer's suggestion to incorporate other PLM-based methods for comparison (including CKB-KBQA [1] and Unseen Entity Handling [2]). We have considered these methods when conducting the experiments. However, we think that they are unsuitable to be used for comparison in our setting based on three major reasons:
> - (1) For the application scenarios, they generally parse the question into the structured query language (e.g., SPARQL) and execute it in a query engine to get answers. In our setting, we only leverage the KG to find the answers.
> - (2) For the adopted core modules, they usually adopt the encoder-decoder architecture (e.g., BART [3], T5 [4]) to produce the structured queries. As a comparison, we usually rely on the PLM+GNN backbone, where the PLM is responsible for questions and relations encoding, and GNN is to perform reasoning on the KG.
> - (3) For the annotated labels, they need both the answer entities and annotated structured queries to learn the parsing procedure, while we only require the answer entities for training.
>
> Considering the above reasons, we didn't incorporate them into the comparison experiments, but we intorduced them in the related work. Besides, we have also tried some other baselines, e.g., KGT5 [5]. However, their performance is not good since our setting is quite different from the original setting of incomplete KG.
>
> [1] Rajarshi Das, Manzil Zaheer, Dung Thai, Ameya Godbole, Ethan Perez, Jay Yoon Lee, Lizhen Tan, Lazaros Polymenakos, and Andrew McCallum. Case-based reasoning for natural language  queries over knowledge bases. EMNLP, 2021.
>
> [2] Xin Huang, Jung-Jae Kim, and Bowei Zou. Unseen entity handling in complex question answering over knowledge base via language generation. EMNLP, 2021.
>
> [3] Mike Lewis, Yinhan Liu, Naman Goyal, Marjan Ghazvininejad, Abdelrahman Mohamed, Omer Levy, Veselin Stoyanov, and Luke Zettlemoyer. BART: denoising sequence-to-sequence pre-training for natural language generation, translation, and comprehension. ACL, 2020.
>
> [4] Colin Raffel, Noam Shazeer, Adam Roberts, Katherine Lee, Sharan Narang, Michael Matena, Yanqi Zhou, Wei Li, and Peter J. Liu. Exploring the limits of transfer learning with a unified text-to-text transformer. JMLR, 2020.
>
> [5] Apoorv Saxena, Adrian Kochsiek, and Rainer Gemulla. Sequence-to-sequence knowledge graph completion and question answering. ACL,2022.

---

> ### Author Response · Authors · 2022-12-06
> **Kindly Reminder for the Discussion**
>
> Dear Reviewer Zm6y,
>
> Thanks for your careful reading of our paper. We have further given a more detailed description of all baselines in Appendix B. And we carefully explained your concerns about the fair comparison during our experiments. We have tried our best to elaborate the unclear points and revised our paper accordingly. We would like to know whether you find our response satisfactory, or if there are more questions that we could clarify. Since the rebuttal stage is coming to an end, we are more than happy to hear your comments and address any of your further concerns during the remaining time.
>
> Best,
>
> Authors

---

> > ### Comment · Reviewer_Zm6y · 2022-12-09
> > **Update the recommendation**
> >
> > Thanks the for authors. The authors have added more baselines, which solve my concerns. I have updated my recommendation.

---

> > > ### Author Response · Authors · 2022-12-10
> > > **Appreciate your new response and updated score**
> > >
> > > Dear Zm6y,
> > >
> > > **We sincerely thank you for the updated comments and score.**
> > >
> > > We appreciate your positive reply and your continued vote to accept the work!
> > >
> > > Best,
> > >
> > > Authors

---

### Official Review · Reviewer_j49Y · 2022-10-27

**Confidence:** 3
**Correctness:** 2
**Technical Novelty And Significance:** 2
**Empirical Novelty And Significance:** 2
**Recommendation:** 6

**Clarity, Quality, Novelty And Reproducibility:**

Overall the paper is clear to read.

The paper should include more details about key choices in the experiments, e.g. size of KB. Also, it makes me worried why the ablated results without any of the introduced technique (in Table 4) can outperform the previous state-of-the-art.

**Strength And Weaknesses:**

The propose model is simple and effective. The results are also very impressive. It can be a go-to solution for multi-hop KBQA. I have some questions about the implementation.

1. Without any of the proposed technique (w/o Pre, Trans), the model already outperforms the previous state-of-the-art. Do you know why this happen?
2. The reasoning and retrieval module share the same input structure and same model architecture, but it seems they do not share the same parameters. This sounds weird to me, and I am not sure why this will lead to improvement in model's performance. How much improvement comes from the pretraining of question-relation matching (see also Fig 3(c))? You should consider emphasizing the pretraining strategy if it leads to a big improvement.
3. What is the size of the KB you use for WebQSP and CWQ? Do you use the full Freebase?


**Summary Of The Paper:**

This paper proposed to learn a multi-hop KBQA model which contains a retrieval and reasoning model that shares the same architecture. Unifying the retrieval and reasoning module let the model share more learned knowledge. Experiments show great performance on three benchmark multi-hop reasoning datasets.

**Summary Of The Review:**

The paper delivered good results in their experiments and the intuition of having a shared reasoning and retrieval model is promising. However, it is hard to tell whether the biggest improvement comes from the advertised techniques. Please consider emphasize on your contributions. Having good numbers itself does not warrant an acceptance of the paper.

---

> ### Author Response · Authors · 2022-11-19
> **Response to the Concerns of the Reviewer [Part 3]**
>
> **4.What is the size of the KB you use for WebQSP and CWQ? Do you use the full Freebase?**
>
> We preprocess the full Freebase following existing work [1] [2]. Specifically, based on the topic entities labeled in original datasets, we reserve the neighborhood subgraph consisting of entities within the four hops of the topic entities for each sample. After this simple preprocessing, the size of KG we used is 147,748,092 for WebQSP and 202,358,414 for CWQ. Based on the preprocessed KG, we conduct the retrieval and reasoning using our proposed approach.
>
> [1] Haitian Sun, Bhuwan Dhingra, Manzil Zaheer, Kathryn Mazaitis, Ruslan Salakhutdinov, and William W. Cohen. Open domain question answering using early fusion of knowledge bases and text. EMNLP, 2018.
>
> [2] Gaole He, Yunshi Lan, Jing Jiang, Wayne Xin Zhao, and Ji-Rong Wen. Improving multi-hop knowledge base question answering by learning intermediate supervision signals. WSDM, 2021.
>
> **5.The paper delivered good results in their experiments and the intuition of having a shared reasoning and retrieval model is promising. However, it is hard to tell whether the biggest improvement comes from the advertised techniques. Please consider emphasize on your contributions. Having good numbers itself does not warrant an acceptance of the paper.**
>
> The unified model architecture and the proposed training strategies together contribute to the major performance improvement, where the former is the foundation of the latter. Specifically, our approaches have introduced three new techniques:
> - (1) We introduce a novel unified model architecture based on PLMs for both retrieval and reasoning stages, which are usually separately treated in existing work.
> - (2) We propose pre-training the PLM module with question-relation matching and sharing it between the two stages, which can support our model to perform better by updating fewer parameters.
> - (3) We propose initializing the reasoning model with the retrieval model to transfer the relevance information between the retrieval and reasoning stages, which is hard to implement in existing work.
>
> In order to test the contribution of each part, we conduct an ablation study to illustrate the effect of different techniques on the final performance. See the following two tables:
>
> \\begin{array} {|l|c|c|}
> \\hline
> & \\textrm{WebQSP (Hits@1)} & \\textrm{CWQ (Hits@1)} \\\\
> \\hline
> \\textrm{SR+NSM} & 68.9 & 50.2 \\\\
> \\hline
> \\textrm{SR+UniKGQA}  & 70.5 & 48.0 \\\\
> \\hline
> \\textrm{UniKGQA+NSM} & 69.1 & 49.2 \\\\
> \\hline
> \\textrm{UniKGQA+UniKGQA}  & 75.1 & 50.7 \\\\
> \\hline
> \\end{array}
>
> From the above table, we can see that:
> - (1) Effectiveness of unified model architecture (last row v.s. other rows): Separately using the retrieval model or reasoning model and then combining it with existing models can not fully utilize the advantages of our unified model architecture. This demonstrates that our unified model indeed effectively shares or transfers useful relevance information across the two stages to boost the final performance.
>
> \\begin{array} {|l|c|c|}
> \\hline
> & \\textrm{WebQSP (Hits@1)} & \\textrm{CWQ (Hits@1)} \\\\
> \\hline
> \\textrm{UniKGQA} & 75.1 & 50.7 \\\\
> \\hline
> \\textrm{w/o Pre}  & 67.3 & 48.1 \\\\
> \\textrm{w/o Trans} & 71.8 & 48.5 \\\\
> \\textrm{w/o Pre, Trans} & 64.1 & 47.5 \\\\
> \\textrm{w QU} & 77.0 & 50.9 \\\\
> \\textrm{w/o Pre, w QU} & 75.4 & 49.2 \\\\
> \\textrm{w/o Trans, w QU} & 75.8 & 49.8 \\\\
> \\textrm{w/o Pre, Trans, w QU} & 72.5 & 48.1 \\\\
> \\hline
> \\end{array}
>
> From the above table, we can see that:
> - (1) Effectiveness of pre-training strategy (w/o Pre v.s. UniKGQA and w/o Pre, w QU v.s. w QU): Once removing the pre-training strategy, the model performance would drop 10.4% (2.1%) in WebQSP and 5.1% (3.3%) in CWQ when fixing (not fixing) the PLM. It indicates that this strategy is an important component of our approach.
> - (2) Effectiveness of initializing the reasoning model with the retrieval model strategy (w/o Trans v.s. UniKGQA and w/o Trans, w QU v.s. w QU):  Once removing the pre-training strategy, the model performance would drop 4.4% (1.1%) in WebQSP and 4.3% (2.2%) in CWQ when fixing (not fixing) the PLM. It indicates that this strategy indeed transfers the relevance information across two stages, which is important for performance improvement.
> - (3) Effectiveness of combining two strategies (w/o Pre, Trans v.s. UniKGQA and w/o Pre, Trans, w QU v.s. w QU): Once removing both two strategies, the model performance would drop 14.6% (5.8%) in WebQSP and 6.3% (5.5%) in CWQ when fixing (not fixing) the PLM. It indicates that by combining these two training strategies, the model can share and transfer the relevance information for multi-hop KGQA between two stages, which leads to the final significant performance improvement.

---

> ### Author Response · Authors · 2022-11-19
> **Response to the Concerns of Reviewer [Part 2]**
>
> **2.The reasoning and retrieval module share the same input structure and same model architecture, but it seems they do not share the same parameters. This sounds weird to me, and I am not sure why this will lead to improvement in model's performance.**
>
> Our unified model architecture contains two groups of parameters: the underlying PLM (group 1), which takes up the majority of the whole parameters (about 90%), and the other parameters (group 2) for matching and propagation. We share the parameters in group 1 for capturing common semantics across two stages and enforce the specificity of each stage by using different copies for the parameters of group 2. In order to transfer the information across stages, we further utilize the parameters of group 2 from the first stage for initializing those for the reasoning stage. Next, we describe the specific techniques and their effect in detail.
> In fact, the reasoning and retrieval models only share the parameters of the PLM (group 1) enhanced by pre-training. Since the basis of the two stages are formulated as evaluating the relevance between a question and a single relation, it is natural to share the underlying PLM, which is the core of our approach.
>
> For other parameters (group 2), as the retrieval and reasoning stages are highly dependent, we use the learned ones from the retrieval model to initialize the reasoning model for transferring useful information. This strategy makes our approach can also capture stage-specific characteristics for achieving good performance.
>
> Specifically, both the retrieval and reasoning models are able to utilize the enhanced PLM to model better the semantic relevance between the question and relation, which is a basic capacity of the two stages. Further, after the retrieval model learns to measure the matching score between the question and nodes in a subgraph, the reasoning model can inherit this ability to perform more accurate reasoning to find the answer entities of the question. Based on these two techniques, the two models sufficiently leverage various kinds of relevance information from the two stages to boost the final performance.
>
> As shown in Figure 3(c), when the fine-tuning epoch of the reasoning model is 0, the reasoning model that fully relies on the parameters of the retrieval model, can achieve comparable performance as competitive baselines (64.8 v.s. 66.4 of GraftNet on WebQSP). Besides, starting from such good initialized parameters, the reasoning model can better learn to accurately find the final answer entities after further fine-tuning, and outperforms previous SOTA in a large margin shown in Table 3 (e.g., 8.1% improvement of hits@1 on WebQSP and 2% improvement of hits@1 on CWQ)
>
> **3.How much improvement comes from the pretraining of question-relation matching (see also Figure 3(c))? You should consider emphasizing the pretraining strategy if it leads to a big improvement.**
>
> The pre-training of question-relation matching contributes a lot to the final performance. As shown in Table 4, without pre-training, the performance drops 2.1% on WebQSP and 3.3% on CWQ. To further investigate its contribution, we add new experiments to study how the convergence speed w.r.t. using pre-training or not, and the final performance changes w.r.t. fixing the PLM or not.
>
> As a newly added experiment, we first plot the performance changing curve of the variant UniKGQA w/o Pre in Figure 3(c). To achieve the same performance on WebQSP (66% Hits@1), our approach, the variation, and the competitive baseline NSM require 1 epoch, 29 epoch, and 45 epoch training, respectively. And finally, our approach achieves the best performance after convergence. It indicates that the pre-training task helps our model convergence faster and better.
>
> In the following table, we conduct ablation experiments to investigate how the pre-training strategy (Pre) affects the performance with or without updating the PLM (QU). Once removing the pre-training strategy, the model performance would drop 10.4% (2.1%) in WebQSP and 5.1% (3.3%) in CWQ when fixing (not fixing) the PLM. It indicates that the pre-training strategy is an important component of our approach. After pre-training, the PLM can be fixed for more efficient parameters optimization during fine-tuning.
>
> \\begin{array} {|l|c|c|}
> \\hline
> & \\textrm{WebQSP (Hits@1)} & \\textrm{CWQ (Hits@1)} \\\\
> \\hline
> \\textrm{UniKGQA} & 75.1 & 50.7 \\\\
> \\hline
>   \\textrm{w QU} & 77.0 & 50.9 \\\\
>   \\textrm{w/o Pre, w QU}  & 75.4 & 49.2 \\\\
>   \\textrm{w/o Pre}  & 67.3 & 48.1 \\\\
> \\hline
> \\end{array}

---

> ### Author Response · Authors · 2022-11-19
> **Response to the Concerns of Reviewer [Part 1]**
>
> Thanks for your insightful suggestions and we have listed our response to your concerns as follows. If you also have any other questions, please feel free to let us know. We will continue to try our best to answer for you.
>
> **1.Without any of the proposed technique (w/o Pre, Trans), the model already outperforms the previous state-of-the-art. Do you know why this happen?**
>
> The main reason is that we adopt a unified PLM-enhanced model architecture for both the retrieval and reasoning stages. Although this idea is intuitive, to our knowledge, there are no studies that explore such a unified architecture for large-scale KBQA. Next, we will explain why this unified architecture leads to a better performance in detail, from the two aspects of (1) higher-quality retrieved subgraphs and (2) stronger reasoning ability:
> - (1) For the retrieval stage, the previous SOTA on WebQSP (i.e., TransferNet) adopts a heuristic Personalized Page Rank-based (PPR-based) algorithm to filter irrelevant entities. Such a way neglects the question semantics, limiting the retrieval accuracy. In our approach, we leverage the unified PLM-enhanced model to retrieve a question-relevant subgraph. As our approach can learn to capture the question semantics via the PLM, it can accurately retrieve a relatively smaller subgraph yet with a higher answer coverage rate. We have shown the superiority of our approach during the retrieval stage in Figure 3(a), where for reaching the answer coverage rate of 90%, our approach only requires retrieving a much smaller subgraph than the PPR-based approach (about 100 v.s. 1400).
> - (2) In addition to the higher-quality retrieval, the PLM-enhanced model architecture also endows our approach with stronger reasoning ability. During the reasoning stage, the previous SOTA on CWQ (i.e., SR+NSM) relies on LSTM to encode the questions and trainable relation embeddings to perform reasoning on the retrieved subgraph. As a comparison, we incorporate a more powerful text encoder, PLM, to generate the representations of the questions and relations. Since PLM has been pre-trained on large-scale general corpus and owns a larger number of parameters, it is able to capture more complex semantics between questions and relations for knowledge reasoning. Therefore, as shown in Figure 3(b), using the same retrieved subgraph, our UniKGQA reasoning model can obtain a better reasoning performance than the SOTA NSM (70% v.s. 68%).

---

> ### Author Response · Authors · 2022-12-05
> **Kindly Reminder for the Discussion**
>
> Dear Reviewer j49Y,
>
> Thanks for your careful reading of our paper. We have tried our best to elaborate the unclear points and revised our paper accordingly. We have added ablation experiments to investigate how the pre-training strategy (Pre) affects the performance with or without updating the PLM. Besides, we have added the ablation study to explore the effect of only adopting the unified model architecture as the reasoning model or the retrieval model to demonstrate the advantages of unifying the two stages. We would like to know whether you find our response satisfactory, or if there are more questions that we could clarify. Since the rebuttal stage is coming to an end, we are more than happy to hear your comments and address any of your further concerns during the remaining time.
>
> Best,
>
> Authors

---

> ### Author Response · Authors · 2022-12-10
> **Appreciate your updated score**
>
> Dear Reviewer j49Y,
>
> **We sincerely thank you for the updated score.**
>
> Thanks for your constructive review. Your review really helped us greatly in improving our paper, and we are truly grateful for your comments. We are very happy to see that you raised the score for our updated revision. And we sincerely appreciate your time and effort in reviewing our paper and reading our comments. If you also have any other questions, please feel free to let us know. We will continue to try our best to answer for you.
>
> Best,
>
> Authors

---

### Author Response · Authors · 2022-11-19
**New Revision of our Manuscript**

We sincerely thank the three reviewers for their insightful and constructive feedback. We have provided a separate response to each reviewer, and also updated the paper following the revision suggestions of the reviewers. We list the main revision content as follows:

- We add the one-shot experiments on MetaQA in Appendix E, and show the experimental results on Table 6. We can see that our approach can consistently outperform competitive baselines in all the three MetaQA datasets.

- We add the ablation study to explore the effect of only adopting the unified model architecture as the reasoning model or the retrieval model in Appendix F, and show the results on Table 7. We can see that the unified model used in the retrieval and reasoning stages simultaneously is indeed the key reason for improvement.

- We conduct the analysis experiments to investigate how the pre-training strategy affects the performance with or without updating the PLM in Appendix G, and show the results on Table 8. We can see that the pre-training strategy is an important component of our approach.

- We fix the mentioned typos, and carefully read and revised the writing of our paper according to the reviewers' suggestions.

- For reproducibility, we upload the code of our approach in the supplementary material.

---

### Decision · Program_Chairs · 2023-01-20

**Decision:**

Accept: poster

**Justification For Why Not Higher Score:**

While the paper presents a simple and effective model, some weaknesses remain, which made the acceptance decision itself difficult. Therefore, I believe it should not be presented as a spotlight paper at ICLR.

**Justification For Why Not Lower Score:**

The paper provides a new solution to a problem with a significant impact. While some weaknesses were pointed out, they concern mostly the writing style and explanations rather than the core contributions of the paper. Therefore, I recommend acceptance.

**Metareview: Summary, Strengths And Weaknesses:**

The paper proposes a new approach for multi-hop question answering over knowledge graphs (KGQA). Most current KGQA systems have a separate retrieval stage, in which relevant parts of the knowledge graph are selected, followed by a reasoning stage aiming at obtaining the correct answer. The proposed approach unifies both stages.

Strengths:
- Simple and effective model
- Clear main contribution
- Strong evaluation results

Weaknesses:
- Doubts about some claims made in the paper
- Some parts of the writing and explanation could be further improved

**Note From Pc:**

if the above contains the word "oral" or "spotlight" please see: "oral" presentation means -> notable-top-5% and "spotlight" means -> notable-top-25%. As stated in our emails, we are disassociating presentation type from AC recommendations

**Summary Of Ac-Reviewer Meeting:**

The meeting summary generally helped to resolve some concerns about the paper and led to an increase in scores that ultimately led to the decision to accept the paper. Basically, two points were discussed:

- The model performance didn’t look plausible at first glance, but after discussion, we resolved this point in favour of the authors.
- Similarly, we also believed the authors did a good job of addressing the request for better descriptions of baselines which was a point of criticism. Thus, this was not seen as a barrier to acceptance anymore.

Two reviewers improved their scores based on the AC reviewer meeting discussions.